# Survival Analysis of Glass Ionomer Cement and Resin-Based Sealant Retention: A 10-Year Follow-Up Study

**DOI:** 10.3390/medicina60050756

**Published:** 2024-05-01

**Authors:** Sandra Petrauskienė, Kristina Saldūnaitė-Mikučionienė, Julija Narbutaitė

**Affiliations:** Department of Preventive and Paediatric Dentistry, Faculty of Odontology, Lithuanian University of Health Sciences, Luksos-Daumanto Str. 6, LT-50106 Kaunas, Lithuania; sandra.petrauskiene@lsmu.lt (S.P.); kristina.saldunaite@lsmu.lt (K.S.-M.)

**Keywords:** caries prevention, resin-based sealant, glass ionomer sealant, fissure sealant, sealant retention

## Abstract

*Background and Objective*: Sealant application is a proven method to prevent occlusal caries; however, long-term studies on this topic are scarce. This study aimed to assess the survival rate and clinical effectiveness of glass ionomer cement (GIC) and resin-based sealants (RBSs) on second permanent molars over a long-term follow-up period. *Materials and methods:* Sixteen patients aged 11–13 years with all four completely erupted permanent second molars were enrolled in the study. All patients attended 1-year and 3-year follow-ups; however, one participant did not respond after 10 years and was excluded from the final analyses. The oral health status evaluation was based on WHO criteria. A total of 32 teeth received an RBS (Clinpro), and a further 32 teeth were sealed with GIC (Fuji IX). The sealant retention was determined according to the Kilpatrick criteria after 1 year, 3 years, and 10 years, respectively. Statistical analysis included a chi-square test, the Kaplan–Meier method, and the Cox proportional hazard model. *Results*: At baseline, seven boys and eight girls participated in the study, with a mean age of 12.3 ± 0.9 years. The 1-year follow-up results revealed that 90% of the RBSs and 43.3% of the GIC sealants were completely retained, and no caries lesions were recorded (*p* = 0.01). The 3-year follow-up results showed that 23.3% of the RBSs and 0% of the GIC sealants demonstrated complete retention (*p* = 0.034). Moreover, 10.0% of the occlusal surfaces in the RBS group and 13.3% of the occlusal surfaces in the GIC group were filled (*p* > 0.05). A total of 6.7% of the RBSs showed complete retention. One-third of the sealed teeth (30.0% of the teeth sealed with RBSs and 36.7% of teeth applied with GIC) were filled after 10 years. The Kaplan–Meier analysis demonstrated a higher survival rate in the RBS group when compared with the GIC over the entire follow-up period (*p* = 0.001). *Conclusions*: Although the survival rate of RBSs was higher than GIC sealants, their effectiveness in preventing fissure caries in permanent second molars did not differ significantly over a 10-year follow-up.

## 1. Introduction

Dental caries prevails as a chronic non-communicable disease in both children and adults, and it also remains the most relevant global oral health burden worldwide [1,2].

Caries prevention remains a challenge due to the high prevalence of disease in recently erupted teeth [3]. Early sealant application after permanent molar eruption allows one to achieve a higher effectiveness in caries prevention [4]. Handelman was the pioneer in terms of analyzing the role of sealant application over incipient dental caries lesions [5]. Studies have shown that teeth to which sealants have been applied may require less extensive dental treatment than teeth without sealants. Furthermore, patients with sealed teeth need less frequent restorative treatment in the future [6]. A systematic review revealed that the application of dental sealants may reduce caries development in permanent posterior teeth from 40% to 6% during a 2-year period [7].

The main indication for dental sealant application is recently erupted permanent teeth with deep pits and fissures [8]. The occlusal surfaces of molars tend to accumulate dental plaque because of pits and fissures, and these may play a role in the inefficient management of occlusal caries [9,10]. Thus, dental sealants are defined as an effective preventive procedure due to serving as a micromechanically bonded protective layer [9,10,11]. Pit and fissure sealants are applied not only for prevention but also for the management of incipient non-cavitated occlusal caries lesions.

The term “pit and fissure sealant” describes a chemically active liquid material that is applied to the occlusal pits and fissures of caries-susceptible teeth. Moreover, it is cured chemically or via light [6]. Pit and fissure sealants were introduced by Buonocore, who presented the acid etching technique and application procedure of pits and fissures to prevent caries in 1955 [12]. Glass ionomer cement (GIC) as a pit and fissure sealant was introduced by McLean and Wilson in 1974 [13].

When considering the chemical composition, there are two main categories of material, including resin-based sealants (RBSs) and glass ionomer cement (GIC). The most common subgroups of sealants, such as polyacid-modified resin sealants, resin-based sealants, resin-modified glass-ionomer sealants, and glass-ionomer sealants, have been evaluated in various studies [14]. Even flowable composites can be used as dental sealants [15]. However, resin-based sealants are defined as the gold standard [16]. A meta-analysis by Lam et al., 2021 proved that RBS application on permanent molars correlates with a reduction in occlusal caries development and with the arrest of caries lesion progression [17]. However, a significant disadvantage of resin-based sealants is their sensitivity to moisture during application procedures [10].

Glass ionomer cement as a dental sealant possesses certain advantages. Glass ionomer cement is a moisture-friendly dental material, and it is capable of releasing fluoride [10,18]. The technique of GIC application is easier to perform when compared to the application of resin-based sealants [19]. GIC enables one to ensure a potential preventive effect of fluoride release even after a high loss of sealed material [19,20]. Thus, this material is an option for partially erupted teeth [19,20]. In considering microleakages, certain brands, such as Fuji IX, are preferred over other GIC sealants [21]. The main drawback of GIC is its relatively poor retention [16,20]. However, it is acceptable to use GIC as a sealant in caries prevention programs [22].

The retention rate of sealants on tooth surfaces is the main indicator of their success [23].

The complete retention of pit and fissure sealants ensures optimal protection [16]. Meanwhile, the sites of dental plaque retention that are caused by the partial loss of sealant may initiate caries development [24]. Early pit and fissure sealant loss is related to salivary contamination during application procedures [25], and it is also dependent on the type of sealant material used [26]. Long-lasting caries prevention is aided by regular dental appointments and the resealing of susceptible tooth surfaces if the previously applied material is deficient [11,27].

Dental sealant is an integrated component of various prevention programs, such as non-operative caries treatment programs (e.g., the Nexö method) [28,29,30]. Subsequently, DMF-S was significantly reduced in the communities that implemented these programs [29,30].

In Lithuania, only 23.61% of schoolchildren have caries-free dentition [31]. The prevalence of caries in permanent molars was 41% among 5- to 6-year-olds [32] and 64.0% among 12-year-old school children in Lithuania [33]. The prevalence of dental caries among 12-year-olds has remained high (88.4% in 1983, 85.5% in 2005, and 70.6% in 2009) over the decades [33,34]. The mean DMF-T score was found to be high (7.9) among 12-year-old Lithuanian children [35]. This situation indicates a high (dental) treatment need, especially non-operative treatments, to control and reduce caries development in Lithuania [32,35].

The sealing of caries-free teeth or enamel-incipient lesions is recommended for patients who are at risk [25,29]. A national program of pit and fissure sealants was initiated in 2004 in Lithuania [36]. The target group for sealant application is children aged 5–13 years with recently erupted permanent molars. Saldūnaitė et al. found that 12-year-old Lithuanian children who had up to two teeth had pit and fissure sealants applied [33].

In considering the effectiveness of occlusal caries prevention, numerous studies have not defined the significant differences between resin-based sealants and glass ionomer cement ones [9].

This study aimed to assess the survival rate and clinical effectiveness of glass ionomer cement (GIC) and resin-based sealants (RBSs) on second permanent molars over a long-term follow-up period.

## 2. Materials and Methods

We conducted this clinical study of sealant retention in permanent second molars at the Clinic of Preventive and Pediatric Dentistry of the Lithuanian University of Health Sciences (LSMU) (Kaunas, Lithuania) during 2004–2016. The study was conducted in line with the principles of the Declaration of Helsinki. The Kaunas Regional Biomedical Research Ethics Committee approved the study protocol (no. 100/2003). The parents of the participants provided their written informed consent for the patients to participate in the study.

### 2.1. Participants

The study targeted patients aged 11–13 years who were attending the Clinic of Preventive and Pediatric Dentistry, LSMU in Kaunas, Lithuania. A total of 16 patients with all four fully erupted permanent second molars were enrolled in the study. The enrollment of participants was carried out from March 2004 until November 2005. Sealant application was performed between April 2004 and December 2005.

All patients attended 1-year and 3-year follow-ups, but one participant did not respond after 10 years and was thus excluded from the final analyses (Figure 1).

#### 2.1.1. Inclusion Criteria

The patient inclusion criteria were the following: 11–13-year-old patients without any known systemic diseases.

The teeth-related inclusion criteria were the following: maxillary and mandibular second permanent molars have completely erupted with sound and intact pits and fissures.

#### 2.1.2. Exclusion Criteria

The patient exclusion criteria were the following: uncooperative patients and patients with mental and/or physical disorders.

The teeth-related exclusion criteria were the following: partially erupted second permanent molars, second permanent molars with proximal caries, and enamel with developmental defects, restorations, or sealants.

### 2.2. Sample Size Calculation

The sample size for the study was calculated using Power Analysis (G-power software™ (Release 3.0) Heinrich Heine University, Düsseldorf, Germany, for Windows). The sample size calculation provided a value of 16 participants (32 teeth per group) with a confidence level of 95% and a type I error (alpha) of 5% when a difference of 43.5% between the complete retention rates of the tested RBS and GIC was assumed at 12 months. A power of 98% indicated that the sample size of 16 participants (32 teeth per group) was sufficient.

### 2.3. Intervention

Prior to the sealant application, clinical examinations were performed and bitewing radiographs were taken to ensure that no caries lesions were present.

The oral health status evaluation was based on WHO criteria [37] under standardized conditions with a plane dental mirror and periodontal (CPI) probe in a dental clinic setting. During follow-ups, oral examinations were performed using the same criteria as the baseline visit. After a dental examination, the scores of the decayed, missed, and filled permanent teeth (DMF-T) index were calculated [38,39].

The presence of dental plaque in each participant was assessed using the Silness–Löe plaque index (PI). Dental plaque was recorded at the gingival areas of the buccal, mesial, distal, and palatal/lingual surfaces of all the teeth. The score for a tooth was counted by adding the scores of each area and dividing by 4. Meanwhile, the score for each patient was counted by adding all the scores of the teeth and then dividing this by the number of teeth, with ratings of 0 to 3 (0—excellent; 0.1–0.9—good; 1.0–1.9—fair; and 2.0–3.0—poor) [38].

Oral examinations, sealant applications, and follow-ups were performed by the same dental hygienist (specialist) (K.S-M).

Overall, sealants were applied to 64 teeth. Resin-based sealants (RBSs) (Clinpro^TM^ Sealant, 3M/ESPE, Saint Paul, MN, USA) were applied to a total of 32 teeth, and 32 teeth were sealed with glass ionomer cement (GIC) sealants (Fuji IX, -Fuji IX, GC, Tokyo, Japan) at baseline. The sealants were applied in accordance with the manufacturers’ instructions (Figure 2). The material for sealant (RBS or GIC) application, with regard to the upper or lower jaw and side (right or left), was selected by the operator. Additionally, moisture control conditions were considered.

Professional oral hygiene was performed, and regular oral hygiene instructions were given to all participants at the beginning and during follow-up appointments.

### 2.4. Outcome Measurement

The integrity of the sealant was assessed clinically during the follow-up appointments after 1 year, 3 years, and 10 years of placement. The retention of the sealant material on each sealed tooth was determined according to the Kilpatrick criteria [40] as follows: 0 indicates complete retention, 1 indicates the loss of 1/3 of the sealant, 2 indicates the loss of 2/3 of the sealant, and 3 indicates the complete loss of the sealant (more than 2/3 of the material). The later scores of sealant retention were regrouped as complete retention (score 0), partial loss (scores 1 and 2), and total loss (score 3). In addition, all the surfaces of the teeth sealed with sealants were evaluated in terms of caries development during follow-up visits. The criteria for evaluation were the following: 0 indicates no caries, and 1 indicates caries (filling) present [41]. Later, “failure” was defined as a partial loss of the sealant or a complete loss of the sealed material and filling.

### 2.5. Statistical Analysis

A statistical analysis was performed using the Statistical Package for Social Sciences (SPSS version 29). A chi-square test served to measure the differences between the assessed sealant groups. The Mann–Whitney U test was used to compare the mean scores of the PI at different time periods. A comparison of different time periods with respect to the mean scores of the DMF-T index and its components (D-T, M-T, and F-T) was performed using the Wilcoxon test. A survival analysis of the different sealant materials was performed with the Kaplan–Meier method and the Cox proportional hazard model. The significance level was set at *p* < 0.05.

## 3. Results

Table 1 shows the demographic characteristics of the study participants, as follows: seven boys (46.7%) and eight girls (53.3%), with a mean age of 12.3 ± 0.9 years at the baseline.

Table 2 shows the oral health status (presence of dental plaque (PI) and severity of dental caries (DMF-T)) of the study participants at the baseline and the 10-year follow-ups. At baseline, the mean score of the PI was 2.53 ± 0.50. The oral hygiene of the participants was slightly improved over the whole follow-up period, with a PI of 2.47 ± 0.62 (*p* = 0.723). Consequently, the mean score of the filled teeth (F-T) significantly increased during the 10-year follow-up period (from 0.93 ± 1.10 to 3.20 ± 2.43; *p* = 0.013). Within the same follow-up period, the dental caries severity (mean DMF-T score) had increased significantly (from 1.13 ± 1.25 to 3.60 ± 2.30; *p* = 0.003).

At baseline, there were 32 permanent second molars sealed with RBSs and 32 sealed with GIC. Table 3 shows that 66.7% of the sealants were completely retained and 6.7% were completely lost after the 1-year follow-up. In considering the type of material, significantly more complete retentions were observed in the RBS group than in the GIC group (90.0% vs. 43.3%, *p* = 0.01). After the 1-year follow-up period, there were no dental caries lesions recorded in the second permanent molars sealed with RBSs or GIC.

After the 3-year follow-ups, the rates of partial retention (36.7%) and the complete loss of sealants (40.0%) were higher when compared with other outcome measurements, such as complete retention and filling. No complete retention of GIC sealants was found. Overall, 11.7% of the sealed occlusal surfaces were filled.

The 10-year follow-ups showed that 6.7% of the RBSs were completely retained, while 36.7% of the RBSs and 53.3% of the GIC sealants were completely lost (*p* = 0.145) (Table 3). Subsequently, one-third of the sealed teeth (30.0% of the teeth sealed with RBSs and 36.7% of teeth applied with GIC) were already filled (*p* = 0.145) (Table 3).

Figure 3 shows the results of the Kaplan–Meier analysis, wherein it demonstrates the higher survival rate that was found in the RBS group when compared with the GIC group over the whole follow-up period (*p* = 0.001). Compared with RBSs, GIC sealants had a higher risk of failure; the hazard ratio (HR) was 1.426 (95% CI, 1.012–2.006, *p* < 0.001). Finally, the HR showed that the risk of caries development after sealant application with GIC was higher than with RBSs, although it did not differ statistically (1.250 (95% CI, 0.585–2.670 and *p* = 0.565)).

Table 4 shows the rate of complete retention for both materials after the 1-year, 3-year, and 10-year follow-up periods with regard to the upper or lower teeth sealed. A higher complete retention rate of both materials was found on the maxillary permanent second molars during all three periods of follow-up (*p* > 0.05).

## 4. Discussion

The main finding of the study was the higher survival rate of resin-based sealants (RBSs) over glass ionomer cement (GIC) sealants when applied to second permanent molars over a long-term follow-up period. The clinical assessments of the sealant retention were performed after 1-year, 3-year, and 10-year periods. The participants’ oral hygiene status was insignificantly improved and remained inadequate over the whole follow-up period. When considering the location, the complete sealant retention of both materials was insignificantly better on the maxillary permanent second molars over the whole period. In the present study, no caries lesion in the sealed second permanent molars was recorded during the 1-year follow-ups. After the 10-year follow-ups, the prevalence of the filled permanent second molars that had been previously applied with RBSs or GIC sealants did not differ significantly.

This study revealed that 90% of the RBSs were completely retained after a one-year follow-up period. Previous studies showed a high complete retention rate for RBSs, which varied from 62.5% to 96.7% after a 1-year follow-up period [3,42,43,44].

After three years, 23.3% of RBSs were completely retained in this study, while other studies reported a higher rate of complete retention of RBSs (which varied from 80.2% to 91.08% after the same period of follow-up [15,45]). The findings revealed that the complete or partial retention rate of GIC sealants was the same as it was found to be in another study performed by Hesse et al. [22].

When considering the 10-year follow-up period, half of the GIC sealants were completely lost, and 36.7% of the teeth applied with GIC sealants were already filled; these findings are in line with the results of another study carried out in Serbia [19]. Wendt et al. found that 65% of second permanent molars applied with resin-based sealants showed complete retention, and 5% of these teeth were found to be filled after 15-year follow-ups [27]. Long-term studies evaluating sealant retention are scarce [2].

This study revealed that the complete sealant retention of both materials was better on maxillary permanent second molars than on mandibular ones (*p* > 0.05). The same pattern of retention was observed in a study carried out in Romania [18]. Meanwhile, a study performed in Croatia showed the opposite results, i.e., better sealant retention on the mandibular molars (*p* < 0.05) [46].

The pit and fissure sealants’ effectiveness was widely investigated in terms of retention, mechanical properties, different bonding, marginal microleakages, infiltration, and the emergence of caries, as detailed below [42,47,48,49,50]. This study employed the conventional acid etching protocol and cotton roll isolation technique during the sealant application procedure. Another study carried out in Lithuania compared the air abrasion and acid etching methods during RBS application and did not find significant differences in the sealant retention rate and caries development after 5-year follow-ups [42]. When considering bonding variations, the etch-and-rinse, multimode universal, and self-etch adhesives were used after acid etching, but the results did not confirm the effectiveness of the adhesive usage during RBS application after short-term follow-ups [47]. Meanwhile, another study revealed that an ethanol-based bonding agent being applied prior to resin-based sealant placement on first permanent molars significantly increased the retention at 12 months [48]. However, with respect to different isolation techniques, various studies have revealed opposite findings. Mattar et al. showed that the chosen type of isolation, such as the Isolite system, rubber dam, or cotton roll isolation, had no impact on the retention rate of pit and fissure sealants [49]; however, another study found that rubber dam isolation ensured better retention of sealants [50].

There are contradictory opinions regarding the clinical effectiveness of GIC sealants in caries prevention. GIC sealants have lower retention rates than RBSs [46,51,52], but the GIC material is more effective in occlusal caries prevention [51]. It has been found that GIC-based sealants applied according to the atraumatic restorative treatment (ART) protocol are effective at preventing dentine caries lesion development [52]. On the other hand, another study concluded that the application of GIC sealants on first permanent molars was not superior in reducing caries lesion development compared to non-applied molars [22]. However, the clinical relevance was focused not on sealant retention, but on the duration of potential caries prevention in previously sealed teeth [18]. Thus, GIC may be a proper material to seal recently erupted permanent molars for children in the high caries risk category when isolation of a tooth is challenging.

Long-term caries prevention strategies focus on non-operative caries treatment and sealant application, only for patients with high caries risk [29,30]. A high mean number of sealed teeth (4.29 [29] and 8.58 [30]) has indicated that oral health education, dietary counseling, personal tooth brushing, professional plaque removal, and fluoride varnish application may be insufficient measures in arresting caries progression for patients with low motivation [29,30]. These various implemented methods are effective in preventing caries development, although one intervention was not found to be more effective (superior) than another (varnish vs. sealant) [53,54].

The data from the National Oral Health Survey in Greece confirmed that sealant placement (low prevalence of sealants) is significantly associated with caries reduction (24%) in the 15-year-old group [55]. The sealant application for the first and second molars covered by the National Health Insurance Service contributed to decreasing dental caries in South Korea [56]. The financial resources of the national program for pit and fissure sealants in Lithuania were used inefficiently [33].

Numerous studies have proved that using merely professional dental measures, such as fluoride varnish or dental sealant applications, is not sufficient in avoiding dental caries development. Thus, oral hygiene instructions and dietary counseling play an essential role in dental caries prevention [57]. Regular supervised toothbrushing may be as efficient as the application of dental sealants in preventing dentin caries lesion development [58]. In considering long-term prognoses, the most favorable results were achieved when dental sealants were integrated as a component in caries prevention programs [59].

### Strength and Limitation

The main strengths of this study need to be considered. The long-term follow-up for the evaluation of sealant retention and caries development may be considered as the main advantage of this study. However, a relatively small sample size was one of the limitations of this study. The split mouth design was not used, although bitewing radiographs were taken prior to the application of sealants. Furthermore, other methods, such as fiber optic transillumination and laser fluorescence, were not employed to detect occlusal caries.

## 5. Conclusions

Although the survival rate of RBSs was higher than that of GIC sealants, their effectiveness in preventing fissure caries in permanent second molars did not differ significantly over a 10-year follow-up period. Glass ionomer sealants could be a good alternative as an initial temporary sealant material due to better moisture tolerance than resin-based sealants when optimal saliva control during sealant application is not feasible.

## Figures and Tables

**Figure 1 medicina-60-00756-f001:**
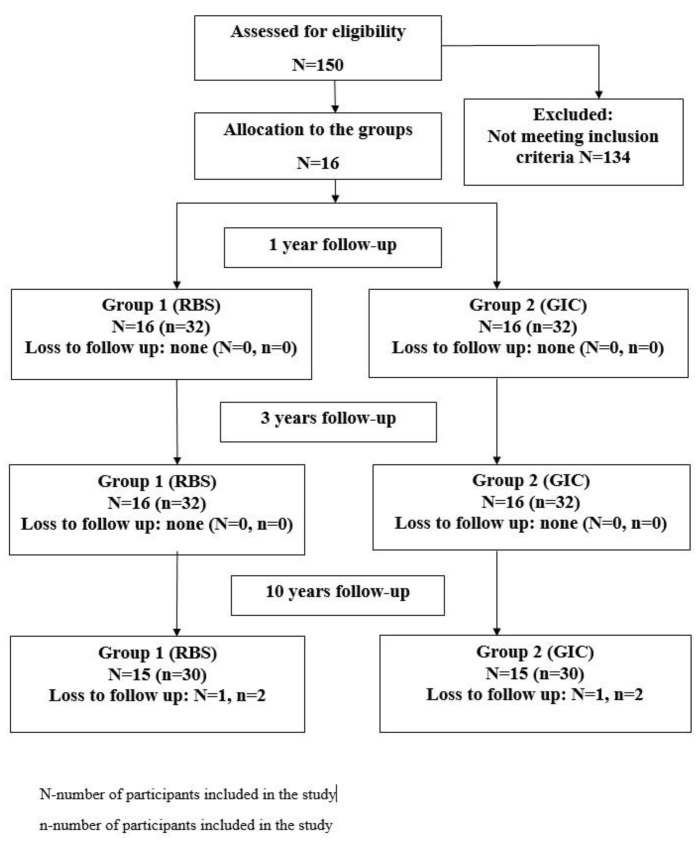
Flowchart diagram of the study sample.

**Figure 2 medicina-60-00756-f002:**
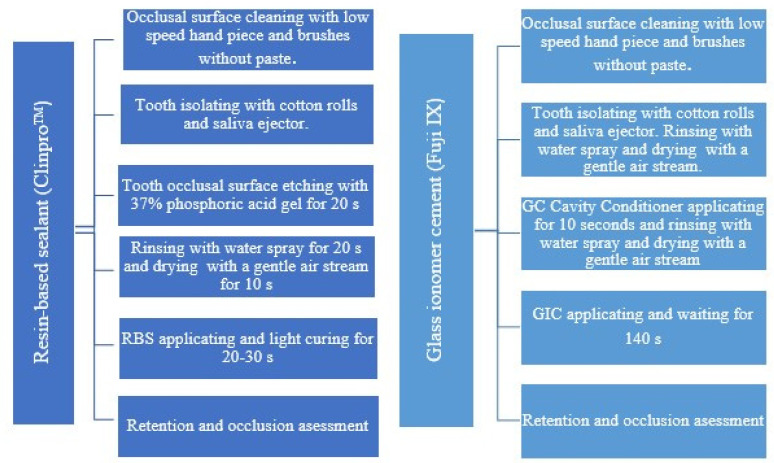
The step-by-step sealant application procedures for both materials (RBSs and GIC).

**Figure 3 medicina-60-00756-f003:**
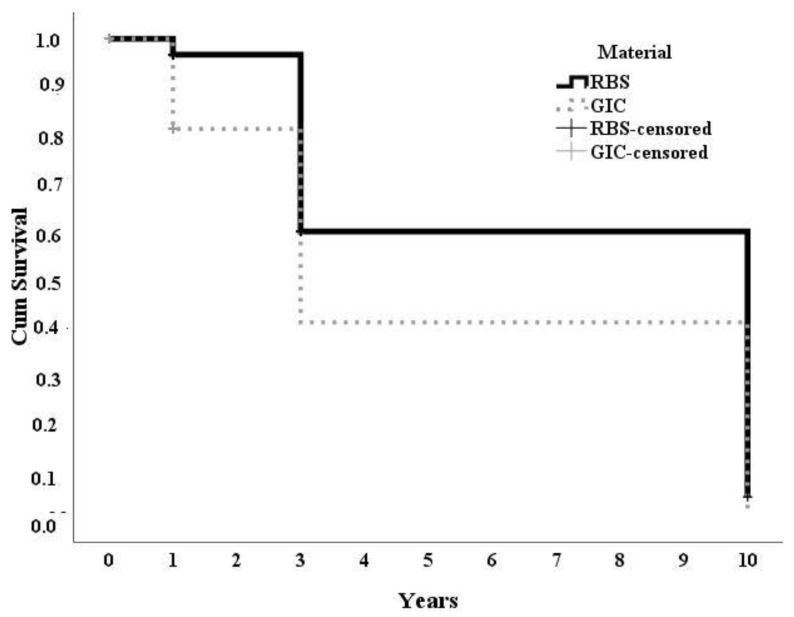
The cumulative survival curves for both types of sealants. “Failure” is defined as the loss of more than one-third of the sealant (via Log rank, *p* = 0.001).

**Table 1 medicina-60-00756-t001:** Characteristics of the participants by year and gender at the baseline.

	N	%
Age (N = 15)		
11 years	4	26.7
12 years	3	20.0
13 years	8	53.3
Total	15	100.0
MS ± SD (years)	12.3 ± 0.9	
Gender (N = 15)		
Boy (male)	7	46.7
Girl (female)	8	53.3
Total	15	100.0

MS ± SD—mean score ± standard deviation.

**Table 2 medicina-60-00756-t002:** Oral status of the participants over the whole follow-up period.

	Follow-Up Period				
Variables	Baseline MS ± SD	1-Year MS ± SD	3-Year MS ± SD	10-year MS ± SD	*p*-Value
Dental caries ^a^					
D-T	0.20 ± 0.41	0.25 ± 0.12	0.31 ± 0.14	0.40 ± 1.30	1.00
M-T	0.0 ± 0.0	0.0 ± 0.0	0.0 ± 0.0	0.0 ± 0.0	1.00
F-T	0.93 ± 1.10 ^b^	1.05 ± 0.19	1.4 ± 0.22	3.20 ± 2.43 ^b^	0.013 ^b^
DMF-T	1.13 ± 1.25 ^b^	1.3 ± 1.15	1.71 ± 1.43	3.60 ± 2.30 ^b^	0.003 ^b^
Presence of dental plaque ^c^					
PI	2.53 ± 0.50	2.73 ± 0.446 ^b^	2.00 ± 0.638 ^b^	2.47 ± 0.62	0.723 < 0.001 ^b^

^a^ Wilcoxon test. ^b^ significant difference between different follow-up periods. ^c^ Mann–Whitney U test. MS ± SD—mean score ± standard deviation.

**Table 3 medicina-60-00756-t003:** Sealant retention characteristics according to the type of material over the whole follow-up period.

Follow-Up Period	Type of Material	Complete Retention, N (%)	Partial Retention, N (%)	Complete Loss, N (%)	Filling, N (%)	Total, N (%)	*p*-Value
1 year	RBS	27 (90.0) ^a^	2 (6.7) ^a^	1 (3.3)	0 (0.0)	30 (100.0)	0.01 ^a^
	GIC	13 (43.3) ^a^	14 (46.7) ^a^	3 (10.0)	0 (0.0)	30 (100.0)	
	Total	40 (66.7)	16 (26.7)	4 (6.6)	0 (0.0)	60 (100.0)	
3 years	RBS	7 (23.3) ^a^	11 (36.7)	9 (30.0)	3 (10.0)	30 (100.0)	0.034 ^a^
	GIC	0 (0.0) ^a^	11 (36.7)	15 (50.0)	4 (13.3)	30 (100.0)	
	Total	7 (11.7)	22 (36.6)	24 (40.0)	7 (11.7)	60 (100.0)	
10 years	RBS	2 (6.6)	8 (26.7)	11 (36.7)	9 (30.0)	30 (100.0)	0.145
	GIC	0 (0.0)	3 (10.0)	16 (53.3)	11 (36.7)	30 (100.0)	
	Total	2 (3.3)	11 (18.4)	27 (45.0)	20 (33.3)	60 (100.0)	

A chi-square test was conducted to compare the results of the sealant retention characteristics and both types of material during the follow-up periods. ^a^ significant difference between RBS and GIC. RBSs—resin-based sealants and GIC—glass ionomer cement.

**Table 4 medicina-60-00756-t004:** Sealant retention characteristics as per the location of sealed teeth over the whole follow-up period.

Sealant Retention	Follow-Up					
	1 Year		3 Years		10 Years	
	RBS, N (%)	GIC, N (%)	RBS, N (%)	GIC, N (%)	RBS, N (%)	GIC, N (%)
Maxillary permanent second molars						
Complete retention	12 (92.3)	8 (47.1)	4 (30.8)	0 (0.0)	2 (15.4)	0 (0.0)
“Failure”	1 (7.7)	9 (52.9)	9 (69.2)	17 (100.0)	11 (84.6)	17 (100.0)
Total	13 (100.0)	17 (100.0)	13 (100.0)	17 (100.0)	13 (100.0)	17 (100.0)
Mandibular permanent second molars						
Complete retention	15 (88.2)	5 (38.5)	3 (17.6)	0 (0.0)	0 (0.0)	0 (0.0)
“Failure”	2 (11.8)	8 (61.5)	14 (82.4)	13 (100.0)	17 (100.0)	13 (100.0)
Total	17 (100.0)	13 (100.0)	17 (100.0)	13 (100.0)	17 (100.0)	13 (100.0)

A chi-square test was conducted to compare the results as per the tooth location and the sealant retention of both types of sealants during the follow-up periods (*p* > 0.05).

## Data Availability

The data presented in this study are available from the corresponding authors.

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
