# Peer review of "Survival Analysis of Glass Ionomer Cement and Resin-Based Sealant Retention: A 10-Year Follow-Up Study"

_medicina, 2024, doi:10.3390/medicina60050756_

Round 1

Reviewer 1 Report

Comments and Suggestions for Authors

The study aimed to evaluate the long-term survival and clinical effectiveness of glass ionomer cement (GIC) and resin-based sealants (RBSs) in preventing occlusal caries on second permanent molars. The research spanned over a decade from 2004 to 2016, providing valuable insights into the durability and efficacy of these preventive measures.

1. One of the significant strengths of this study is its extensive duration, covering a span of 12 years. This prolonged observation period allows for a comprehensive assessment of the longevity and performance of the sealants over time, providing valuable insights for clinical practice.

2. The study employed a thorough methodology, including clinical examination, radiographic assessment, and strict adherence to ethical standards. This comprehensive approach enhances the credibility of the findings and ensures robust data collection.

3. Rigorous statistical analysis, utilizing the Chi-square test, Kaplan–Meier method, and Cox proportional hazard model, adds strength to the study's conclusions. These analytical tools provide a robust framework for evaluating the differences between GIC and RBS in terms of retention and efficacy.

4. By comparing two commonly used sealant materials, GIC and RBS, the study offers valuable insights into their relative effectiveness in preventing fissure caries. Such comparative analyses are essential for guiding clinical decision-making and selecting the most appropriate sealant for individual patients.

5. The study's sample size is relatively small, with only 16 patients included in the analysis. A larger sample would have increased the statistical power and reliability of the findings, allowing for more robust conclusions.

6. One participant was excluded from the final analysis due to non-response after ten years, potentially introducing bias into the results. Strategies to mitigate dropout rates, such as regular follow-up reminders and incentives, could have strengthened the study's validity.

7. The study was conducted in a specific clinical setting in Lithuania, which may limit its generalizability to broader populations or different healthcare contexts. Future research should aim to replicate these findings in diverse settings to ensure their applicability across various demographics and regions.

8. While RBSs demonstrated higher retention rates compared to GIC sealants, the absolute retention percentages were suboptimal for both materials, particularly over the long-term follow-up. This raises questions about the overall efficacy of sealant application in preventing occlusal caries beyond the short term.

9. Please check again all references, if they match exactly with their number.

Overall, the study provides valuable insights into the long-term effectiveness of GIC and RBS sealants in preventing occlusal caries on second permanent molars. While the survival rate of RBSs appears higher than GIC sealants, both materials exhibit limitations in terms of long-term retention and efficacy. Future research endeavors should address these shortcomings and explore innovative approaches to enhance the durability and effectiveness of sealant applications in pediatric dentistry.

Author Response

Our responses to the Reviewer‘s comments

We thank the reviewer for the constructive comments. Please find enclosed our responses highlighted in red.

The study aimed to evaluate the long-term survival and clinical effectiveness of glass ionomer cement (GIC) and resin-based sealants (RBSs) in preventing occlusal caries on second permanent molars. The research spanned over a decade from 2004 to 2016, providing valuable insights into the durability and efficacy of these preventive measures.

  1. One of the significant strengths of this study is its extensive duration, covering a span of 12 years. This prolonged observation period allows for a comprehensive assessment of the longevity and performance of the sealants over time, providing valuable insights for clinical practice.

We thank Reviewer for the comment.

  1. The study employed a thorough methodology, including clinical examination, radiographic assessment, and strict adherence to ethical standards. This comprehensive approach enhances the credibility of the findings and ensures robust data collection.

We thank Reviewer for the comment.

  1. Rigorous statistical analysis, utilizing the Chi-square test, Kaplan–Meier method, and Cox proportional hazard model, adds strength to the study's conclusions. These analytical tools provide a robust framework for evaluating the differences between GIC and RBS in terms of retention and efficacy.

We thank Reviewer for the comment.

  1. By comparing two commonly used sealant materials, GIC and RBS, the study offers valuable insights into their relative effectiveness in preventing fissure caries. Such comparative analyses are essential for guiding clinical decision-making and selecting the most appropriate sealant for individual patients.

We thank Reviewer for the comment.

  1. The study's sample size is relatively small, with only 16 patients included in the analysis. A larger sample would have increased the statistical power and reliability of the findings, allowing for more robust conclusions.

A power of 98% indicated that the sample size of 16 participants (32 teeth per group) was sufficient at 12-month follow-up period (p. 4).

  1. One participant was excluded from the final analysis due to non-response after ten years, potentially introducing bias into the results. Strategies to mitigate dropout rates, such as regular follow-up reminders and incentives, could have strengthened the study's validity.

We thank Reviewer for the comment.

  1. The study was conducted in a specific clinical setting in Lithuania, which may limit its generalizability to broader populations or different healthcare contexts. Future research should aim to replicate these findings in diverse settings to ensure their applicability across various demographics and regions.

We thank Reviewer for the comment. The Clinic of Preventive and Pediatric Dentistry where study was performed is typical dental clinic facilitated as primary care center dental clinic in the Lithuania.

  1. While RBSs demonstrated higher retention rates compared to GIC sealants, the absolute retention percentages were suboptimal for both materials, particularly over the long-term follow-up. This raises questions about the overall efficacy of sealant application in preventing occlusal caries beyond the short term.

Caries prevention focus on non-operative caries treatment and sealant application for patients with high caries risk. Dental sealants are defined as an effective preventive procedure because of a micromechanically bonded protective layer on recently erupted tooth occlusal surface. Since dental caries is a behavioural disease with well known risk factors: dental plaque and high content of carbohydrates.  So in the long term perspective those patients’ behaviuoral factors might be more important in caries development than mechanical sealant protection of vulnerable pits and fissures in recently erupted teeth.

  1. Please check again all references, if they match exactly with their number.

Amended as suggested.

  1. Overall, the study provides valuable insights into the long-term effectiveness of GIC and RBS sealants in preventing occlusal caries on second permanent molars. While the survival rate of RBSs appears higher than GIC sealants, both materials exhibit limitations in terms of long-term retention and efficacy. Future research endeavors should address these shortcomings and explore innovative approaches to enhance the durability and effectiveness of sealant applications in pediatric dentistry.

We thank Reviewer for the comment.

Reviewer 2 Report

Comments and Suggestions for Authors

Dear Authors,

It is an interesting topic. However, the article needs to be changed before publication:

Why do you use Fuji IX as a sealant? Recommended indications for Fuji IX are:

1. Class I and II restorations in deciduous teeth.

2. Non-load bearing Class I and Class II restorations in permanent teeth.

3. Intermediate restorative and base material for heavy-stress situations in Class I and Class II cavities using the sandwich laminate technique.

4. Class V and root surface restorations.

5. Core build-up. 

Fuji Triage is a glass ionomer indicated for pit and fissure sealants.

Also, before applying Fuji IX the surface needs to be prepared with CONDITIONER (GC).

Do you apply a final coat or varnish for finishing? 

If you could not get Triage, explain the reason for Fuji IX.

You cite and compare research (e.g. references 10,14,18) where Fuji Triage was used, and e.g. reference 21 is a study where Fuji IX was used but in vitro.

I noticed that the study under reference number 22 used Fuji IX, but it would be good if you had more comparisons with the same glass ionomer material.

Abbreviations are missing.

Comments on the Quality of English Language

Minor editing of English language required

Author Response

Our responses to the Reviewer‘s comments

We thank the reviewer for constructive comments. Please find enclosed our responses highlighted in red.

  1. Why do you use Fuji IX as a sealant?

Recommended indications for Fuji IX are:

  1. Class I and II restorations in deciduous teeth.
  2. Non-load bearing Class I and Class II restorations in permanent teeth.
  3. Intermediate restorative and base material for heavy-stress situations in Class I and Class II cavities using the sandwich laminate technique.
  4. Class V and root surface restorations.
  5. Core build-up. 

GC Fuji IX was chosen as a glass ionomer cement sealant (adhesive restorative material) for caries prevention in this clinical study.

  1. Fuji Triage is a glass ionomer indicated for pit and fissure sealants. Do you apply a final coat or varnish for finishing? If you could not get Triage, explain the reason for Fuji IX.

The following materials Fuji Triage and G-Coat plus were launched to the market later than this study was planed in year 2003 and conducted in year 2004-2005.

  1. Also, before applying Fuji IX the surface needs to be prepared with CONDITIONER (GC).

During the application technique, GC Cavity conditionier was applied according to the manufacturer‘s instructions. The scheme illustrating  the step-by-step sealant application procedures for both materials (RBSs and GIC) was amended as recommended (Figure 2; p.5).

  1. You cite and compare research (e.g. references 10,14,18) where Fuji Triage was used, and e.g. reference 21 is a study where Fuji IX was used but in vitro.

References 10, 14, 18 and 21 are cited in the Introduction by aiming to present and discuss the advantages and disadvantages of glass ionomer cement used as a sealant.

  1. I noticed that the study under reference number 22 used Fuji IX, but it would be good if you had more comparisons with the same glass ionomer material.

We thank Reviewer for the comment. We were searching for the more studies where Fuji IX was used as a sealant, unfortunately were not able to find more than this one, as a Ref#22.

  1. Abbreviations are missing.

Amended as suggested (p. 2).

Reviewer 3 Report

Comments and Suggestions for Authors

This study aimed to assess the survival rate and clinical effectiveness of glass ionomer cement and resin-based sealants on second permanent molars over a one, three and 10 year follow-up period.

The topic is interesting, even though it is already well-known and widely discussed.

The authors have written an interesting manuscript that meets all the criteria of a scientific work and are commended for the same.

The study is properly written, but there are ambiguities that need to be corrected.

1. Shorten the abstract to 250 words. Delete unnecessary parts (e.g. A clinical study was performed in a clinic for the preventive and pediatric dentistry of the Lithuanian University of Health Sciences (LSMU) (Kaunas, Lithuania) from 2004 until 2016. The study was approved by the Kaunas Regional Biomedical Research Ethics Committee (no. 100/2003).; however, one participant did not respond after 10 years and was excluded from the final analyses.; solation was ensured with cotton rolls and a saliva ejector etc.)

2. Table 1 lists the demographic data for all examined periods if they differ. Also do the same in Table 2.

3. State the reasons and advantages of this study.

Author Response

Our responses to the Reviewer‘s comments

We thank the reviewer for the constructive comments. Please find enclosed our responses highlighted in red.

  1. Shorten the abstract to 250 words. Delete unnecessary parts (e.g. A clinical study was performed in a clinic for the preventive and pediatric dentistry of the Lithuanian University of Health Sciences (LSMU) (Kaunas, Lithuania) from 2004 until 2016. The study was approved by the Kaunas Regional Biomedical Research Ethics Committee (no. 100/2003).; however, one participant did not respond after 10 years and was excluded from the final analyses.; solation was ensured with cotton rolls and a saliva ejector etc.)

We thank Reviewer for the comment and suggestion. We have deleted unnecesary parts as suggested. The abstract was shortened according to the instructions for authors of Medicina journal („The abstract should be a total of about 300 words maximum“).

  1. Table 1 lists the demographic data for all examined periods if they differ. Also do the same in Table 2.

Table 1 presented the demographic data (participants‘ age and gender) at the baseline, while Table 2 presented oral status of participants over the whole follow-up period. Table 1 was amended as suggested to specify the period (p. 6).

  1. State the reasons and advantages of this study.

There is a limited data of carried out studies evaluating the clinical effectiveness of dental sealants in the Lithuania. Thus, this study aimed to assess the survival rate and clinical effectiveness of glass ionomer cement and resin-based sealants on permanent molars over a long-term follow-up period.

The main advantage of this study was the long-term follow-up for the evaluation of sealant retention and caries development.

Round 2

Reviewer 3 Report

Comments and Suggestions for Authors

The authors obviously do not understand what is being asked of them.

It is stated that the follow-up was done after one, three and 10 years, however, in table 2, the oral status after 1 and 3 years is not stated - WHY? Please do corrections!

Table 3 shows the retention of the sealants were completely after the 1-year follow-up, where the data are after 3 years and 10 years.

The results are not well presented, so this kind of manuscript cannot be accepted at the moment.

Author Response

Our responses to the Reviewer‘s comments

We thank the reviewer for constructive comments. Please find enclosed our responses to highlighted in red.

Comments and Suggestions for Authors

The authors obviously do not understand what is being asked of them.

  1. It is stated that the follow-up was done after one, three and 10 years, however, in table 2, the oral status after 1 and 3 years is not stated - WHY? Please do corrections!

We thank Reviewer for the clarification of the question. Table 2 now is amended as suggested and presented in the manuscript (p. 6).

  1. Table 3 shows the retention of the sealants were completely after the 1-year follow-up, where the data are after 3 years and 10 years.

Table 3 presents prevalence of complete retention, partial retention, complete loss and filling of both RBS and GIC material after 1 year, 3 years and 10 years follow-up (p. 7). The Table 3 in the revised and submitted (Round 1) manuscript was presented as is presented below.

Table 3. Sealant retention characteristics according to the type of material over the whole follow-up period

Follow-Up Period

Type

of Material

Complete Retention, N (%)

Partial Retention,

 N (%) 

Complete Loss, N (%)

Filling, N (%)

Total, N (%)

p-Value

1 year

RBS

27 (90.0)a

2 (6.7)a

1 (3.3)

0 (0.0)

30 (100.0)

0.01a

GIC

13 (43.3)a

14 (46.7)a

3 (10.0)

0 (0.0)

30 (100.0)

Total

40 (66.7)

16 (26.7)

4 (6.6)

0 (0.0)

60 (100.0)

3 years

RBS

7 (23.3)a

11 (36.7)

9 (30.0)

3 (10.0)

30 (100.0)

0.034a

GIC

0 (0.0)a

11 (36.7)

15 (50.0)

4 (13.3)

30 (100.0)

Total

7 (11.7)

22 (36.6)

24 (40.0)

7 (11.7)

60 (100.0)

10 years

RBS

2 (6.6)

8 (26.7)

11 (36.7)

9 (30.0)

30 (100.0)

0.145

GIC

0 (0.0)

3 (10.0)

16 (53.3)

11 (36.7)

30 (100.0)

Total

2 (3.3)

11 (18.4)

27 (45.0)

20 (33.3)

60 (100.0)

A Chi-square test was conducted to compare the results of the sealant retention characteristics and both types of material during the follow-up periods. a significant difference between RBS and GIC. RBSs—resin-based sealants and GIC—glass ionomer cement.

  1. The results are not well presented, so this kind of manuscript cannot be accepted at the moment.

We thank again Reviewer for the comments. Hope that suggested amendments in the results are presented in acceptable form.